# Baton: Enhancing Batch-wise Inference Efficiency for Large Language Models via Dynamic Re-batching

## Abstract

The advanced capabilities of Large Language Models (LLMs) have inspired the development of various interactive web services or applications, such as ChatGPT, which offer query inference services for users. Unlike traditional DNN model, the inference of LLM entails different iterations of forward computation for different queries, which result in efficiency challenges for existing run-to-completion batch-wise inference. Hence, some methods refine batch-wise inference to iteration-level by duplicating all nonlinear layers of LLM. However, such the approach not only increases resource usage but also introduced idle computations to the batch due to the prefilling of newly added queries.

Therefore, we propose Baton, an efficient batch-wise LLM inference scheme by dynamically adjusting processing batch, which can achieve near-zero idle computations without incurring additional resource consumption. To do so, Baton 1) shapes the vectors involved in the inference of the newly inserted query and processing batch to align dimensions and generates a new attention mask based on vector shaping to ensure inference correctness, which enable query inserting without consuming additional resource; 2) embeds prefilled *Keys* and *Values* of the new query into the *KV_Cache* of the processing batch by leveraging the prefilling and decoding separation mechanism, eliminating idle computations to the batch introduced by the prefilling process of the new query. Experimental results show that compared to the state-of-the-art solution Orca, Baton outperforms improves query processing by up to 1.75×.

## Keywords

LLM, inference serving, query scheduling

## 1 Introduction

The superior abilities of Large Language Models (LLMs) provide new possibilities for various fields from natural language processing to science researches, which prompts the development of LLM-based web services and applications [1–4]. The ChatGPT from Openai is a notable LLM-based application for users [5], followed closely by Copilot from Microsoft, Gemini from Google, ERNIE-Bot from Baidu, Qwen from Alibaba, Kimi-Chat from Moonshot [6–10], etc., which all can process the queries proposed by users in a interactive conversation manner. These applications essentially provide LLM inference services from a computational perspective, processing user queries by inputting them into the deployed model and returning generated outputs [11]. To fully exploit the parallel computing capabilities of GPUs, this inference process is typically performed in batch-wise. This means the model takes multiple queries stacked along the tensor dimension as input, and generates corresponding result for each query embedded in the output tensor.

The workflow of LLM inference differs significantly from that of conventional Deep Neural Network (DNN) models. The inference of DNN models typically involves a single forward computation to produce the entire output, which naturally aligns with batch-wise processing [12]. For instance, in image classification tasks, an RNN-based model can get classify results of a batch of images via a single forward computation [13]. In contrast, LLM inference follows a unique autoregressive pattern, requiring multiple forward computations. Each forward computation, i.e., an iteration, will generate a new token [14] for each query. Moreover, the inference process can be divided into two phases: prefilling and decoding [15, 16]. In the prefilling phase, the entire query tensor is embedded, and the corresponding *Keys* and *Values* tensors are computed and cached, concluding with the generation of the first new token. The decoding phase then iteratively generates a new token for each query based on last token and the *KV-Cache* [17]. In the meantime, the *KV-Cache* will also be updated in each iteration. The decoding phase continues until the model outputs an end-of-inference symbol (⟨EOS⟩) or the sequence reaches the setting maximum length.

Token is the basic unit that constitutes the final answer sequence to a query. Consequently, the number of inference iteration required depends on the answer sequence length of each query. In existing run-to-completion inference frameworks, inference computation of a batch continues until all queries are completed, even if different queries require varying numbers of iterations. Queries that complete earlier continue to be processed alongside incomplete queries, producing only the ⟨EOS⟩. This unnecessary resource consumption and computation, without generating meaningful tokens, is termed as idle computation, leading to resource underutilization and reducing the efficiency of inference services. Parallel computing on GPUs requires the dimensions of corresponding tensors to be aligned or compatible for multiplication. The autoregressive characteristic of LLMs makes the tensor dimensions involved in inference, such as the *KV-Cache*, to increase with each iteration. Therefore, replenishing a new query directly to current processing batch is impracticable [18], which makes it difficult to utilize aforementioned idle resources.

Orca [19] is the state-of-the-art solution for LLM inference, which refines the scheduling granularity from batch-level to iteration-level and detours dimensional discrepancies caused by inserting new queries via modifying the model structure. The method concatenates queries tensors into a single one-dimensional tensor for all linear computations and replicates the self-attention modules for each query to perform nonlinear computations. On the basis of Orca, FastServe [20] further introduces a query scheduler to support proactive inference task scheduling. However, such the Orca series method faces the following two challenges: 1) the parameters of self-attention take up a large part of the LLM [21–23], replicating manner makes the model consume more resources; 2) the synchronization requirement of the linear layer computation makes the Prefilling of the new query choke the decoding of the other queries of the batch [15, 24].

Happy Birthday

Prefilling (1st iteration)

Decoding (2nd iteration)

Decoding (3rd iteration)

Happy Birthday To You

**Figure 1: Example of** $Q$, $K$, **and** $V$ **calculations during KV-Cache-based inference in a text generation task.**

To this end, we propose BATON in this paper, a efficient batch-wise LLM inference scheme that allows removing completed queries from and inserting new queries to the current processing batch dynamically with near-zero idle computations, like the relay race manner. BATON offers a non-invasive and generic solution applicable to all existing LLMs that utilize the *KV-Cache* policy, which (1) shapes the vectors of new and processing queries by padding operations to align dimensions and creates corresponding masks to ensure the correctness of subsequent inference iterations; (2) embeds the *Keys* and *Values* of new query into the cached *KV* tensors without introducing extra paddings required for dimension alignment by decoupling the prefilling and decoding phases, thereby avoiding additional idle computations, referred to as "bubbles" in the paper. Moreover, BATON instinctively supports query inference interruption and recovery, enabling preemptive query scheduling and flexible batch size scaling during inference.

The contributions of this paper are summarized as follows:

- We designed a tensor shaping and embedding strategy to achieve query-level seamless batch-wise LLM inference, supporting query exits from and inserts to the current processing batch.
- We designed a tensor alignment policy based on P&D decoupling, avoiding the resource and computational bubbles introduced by the embedding process and freeing the implicit constraints of batch composing.
- We conducted extensive experiments with several representative LLMs, and the results show that BATON outperforms the state-of-the-art solution w.r.t. query processing throughput by 1.29-1.75×.

## 2 Preliminary and Motivation

In this section, we firstly introduce some preliminaries about LLM to facilitate the introduction of BATON more explicitly to follow; and

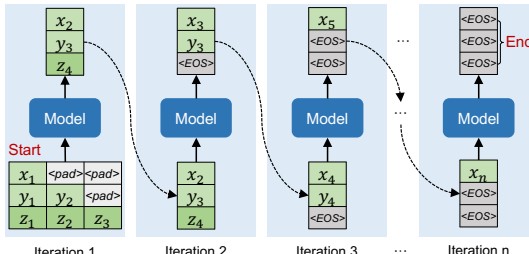

**Figure 2: Batch-wise inference**

then present the motivation of BATON based on practical serving demands.

### 2.1 Preliminaries

In the following, we present preliminaries of LLMs in terms of (1) general architecture, (2) *KV-Cache*-based inference iterations, and (3) workflow of bath-wise inference.

**General architecture.** Most of current open-source LLMs, e.g., GPT and Llama, are mainly based on the Transformer structure or its variations [25, 26]. The key part of Transformer is self-attention mechanism [27]. For self-attention calculation, the input sequence $X$ is first linearly transformed by the trained weight matrices $W^Q$, $W^K$, and $W^V$, generating the Query ($Q$), Key ($K$), and Value ($V$) vectors. The relationship strength between elements of $X$ is evaluated by calculating the dot product between $Q$ and $K$, and then converted into attention weights through scaling and softmax operations. Finally, these attention weights are multiplied by $V$, and through weighted summation, an output is generated.

This process enables the model to integrate global information and dynamically adjust the representation of each element to capture complex dependencies and contextual information in the sequence. Typically, LLMs consist of multiple Transformer layers, between which are set some Feed Forward Networks (FFNs) or Multi-Layer Perceptrons (MLPs).

**KV-Cache-based inference iterations.** Each forward calculation of LLM generates a token based on the input sequence $X$, which is the basic unit of the final response sentence, therefore, it is necessary to append the output token to $X$ iteratively until the complete response sentence is generated. This is the autoregressive nature of LLM and the inference will end when the model outputs the end-of-inference symbol ($\langle EOS \rangle$) or the sequence length reaches the set threshold.

Taking text generation tasks as an example, suppose the initial input sentence is "Happy Birthday", the model will generate "To" in first iteration; and "Happy Birthday" will be input into the model again for the second iteration inference. Based on the aforementioned self-attention description, in a strawman way, it is necessary to calculate the attention weights among "Happy", "Birthday", and "To". However, the attention weights between "Happy" and "Birthday" has been computed in the first iteration. Therefore, these repeated calculations can be avoided by caching previously calculated $K$ and $V$ values and reusing them, which is known as *KV-Cache* [28]. As shown in the Figure 1, the $K$ and $V$ of the first iteration are cached for the second iteration, it is possible to output "You" by calculating "To" related values; it is similarly to calculate the "You"

related values based on the last *KV-Cache* in third iteration. Moreover, given the *KV-Cache* mechanism, only the first iteration—often referred to as the prefilling phase—inputs the current complete query sentence, whereas subsequent iterations, known as the decoding phase, the latest output token is enough as the input.

**Batch-wise inference.** It is necessary to align the dimensions of the involved vectors during batch-wise processing by GPUs. Therefore, as shown in Figure 2, batch-wise LLM inference needs to align the vector lengths of all query sentences of the batch for the prefilling phase, i.e., padding the shorter query by the predefined padding symbol. To avoid the impact of padding symbols on the computation, a 0-1 attention mask vector will be generated for each query to identify the initial tokens and padding symbols.

During inference, if any query in the batch is completed, represented by the output of the ⟨EOS⟩, the model will continue to generate ⟨EOS⟩ directly for that query in subsequent iterations to align vectors dimensions. The inference for this batch continues until all queries in the batch have completed.

## 2.2 Motivation

Regarding LLM inference, how to combine batch is a major area of research, the core of which is dedicated to reducing the idle computation generated by misalignment occurring in batch-wise inference process. For example, avoiding queries with huge difference in initial sentence lengths to compose a batch. Although the computation of the prefilling phase can be performed by padding operation, the computation overhead of prefilling for the whole batch is determined by the longest query, i.e., an excessively long query would stall the computation of the other queries. Since the length of query is explicit, the idle computation in the prefilling phase can be reduced by combining queries with similar length into a batch. However, the inference process of LLM is autoregressive, which means that the rounds of iteration in the decoding phase of each query are uncertain or hardly evaluated, so the batch-wise inference of LLM still faces the issue of idle computation.

**Opportunity: Query-level seamless batch-wise inference.** In the inference process, a query that has completed still occupies GPU resources due to idle computations, repeatedly outputting duplicate ⟨EOS⟩ tokens in each iteration. This issue can be addressed by replenishing a new query seamlessly into the current processing batch after one query completes, akin to a relay race. However, implementing this replenishable feature presents several challenges.

**Challenge 1: Misaligned vector dimension.** Parallel computation on GPUs requires that the dimensions of the vectors for all queries of the batch remain consistent. However, dimension of these vectors changes iteratively during inference, as the illustrative example in Figure 1. Consequently, it is impossible to directly insert a new query into the current processing batch. The SOTA methods, Orca and its derivatives [19, 29, 30], detour dimension misalignment by splicing all vectors into a one-dimension vector for linear layer computing and replicating self-attention networks for each query to enable individual computing. Nevertheless, this approach introduces additional model parameters that consume extra GPU resources proportional to the batch size. Moreover, this replication policy underutilizes the parallel computing capabilities of the GPU.

**Challenge 2: Redundant prefilling.** The currently processed queries are all in the decoding phase, whereas newly inserted queries must first undergo prefilling. This difference in phases causes existing queries to engage in unnecessary idle computations (redundant prefilling) along with the prefilling processing of the new query, even though they only need to compute the most recent single token. For Orca [19], the linear computation of each transformer layer requires outputs of all self-attention replica, this synchronization actually remains the redundant prefilling issue. In spite of employing the strategy of prioritizing the insertion of shorter queries, it is not possible to completely avoid redundant prefilling of existing queries when introducing new ones.

## 3 Baton Overview

### 3.1 Desired properties

To improve the performance of LLM inference serving, the batch-wise processing by leveraging the parallel capabilities of GPUs is advantageous. However, the autoregressive feature of LLMs poses challenges to the current employed run-to-completion batch-wise inference approach. Our goal is to provide an LLM inference serving that meet the following two requirements.

**Query-level continues inference.** The run-to-completion batch-wise will cause longer waiting time for others queries due to the LLM autoregressive. Instead, batch-wise inference should support the insertion of new query after any query is completed, exhibiting the continuity of inference at the query granularity.

**Resource and Computation efficient.** GPU memory resource is valuable especially in LLM scenes, therefore, it is allowed to introduce no or very rare additional resource and calculation overheads in meeting the last requirement.

### 3.2 Baton solutions

For the aforementioned two challenges and desired properties, Baton provides two solutions respectively.

**Solution 1: Vector shaping.** Given the padding operation and attention mask mechanism supported by existing LLMs by default, Baton proposes a vector shaping and embedding strategy, the core idea of which is to align the dimensions of the queries in the processing batch and the newly inserted query by padding operation, and generate a new attention mask based on the padding status to guarantee the correctness of the subsequent inference iterations.

**Solution 2: Vector embedding.** Baton solves this problem by decoupling the prefilling and decoding phases, i.e., all query can be composed into a batch using the similarity of length principle and complete the prefilling phase. When a new query is inserted into the processing batch, the existing queries do not need to be aligned with it by padding, and the entire batch can seamlessly execute the subsequent decoding phase. By ingenious implementation, it allows parallel computation with decoupled prefilling and decoding and enables prefetchable GPU&CPU hybrid *KV-Cache* storing.

## 4 Baton Design

In this section, we first describe the vector shaping scheme for query inserting to achieve query-level continues batch-wise inference. Then we present the vector embedding strategy to zeroing the

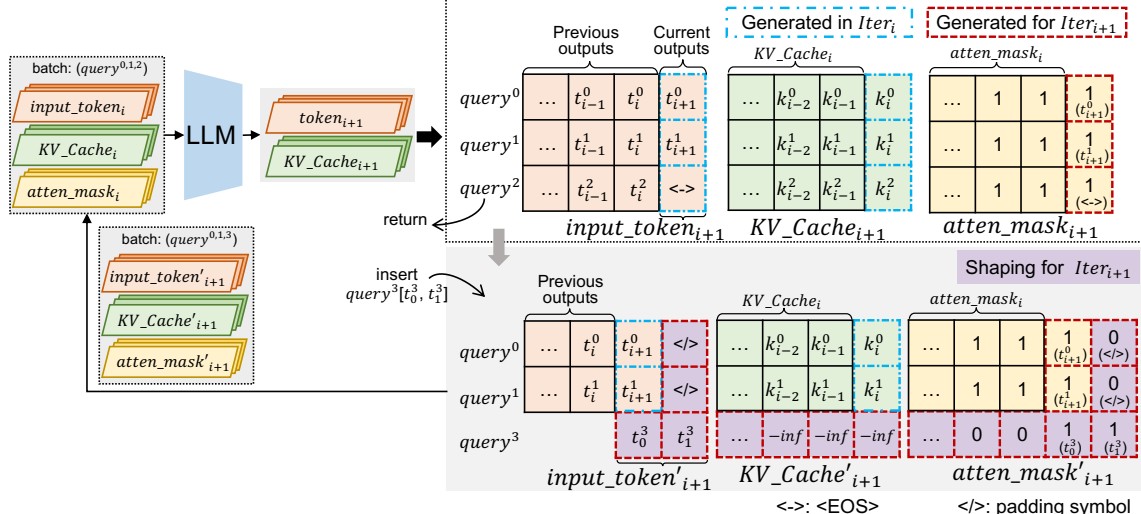

Figure 3: BATON: an illustrative example of query inserting

bubbles introduced by query inserting. At last, we show additional functions of BATON that can further improve the inference serving system.

## 4.1 Vector Shaping

In the LLMs inference that employs *KV-Cache* mechanism generally involves three primary variables: *input_token*, *attention_mask*, and *KV_Cache*. Initially, for the first iteration inference, *input_token* contains all tokens for each query of the batch, which will be aligned by the padding operation. That is, *input_token* is the dimension of the $batch\_size \times l_{max}$, where $l_{max} = max(query_i)$ is the length of the longest query. To facilitate the description, the embedding dimension of tokenizer is omitted in this paper as it is fixed for all tokens, and we mainly focus on the dimension of token sequence length. The original and padding tokens are distinguished by a 0-1 vector known as *attention_mask*, whose dimension keeps consistent with that of *input_token*. And *KV_Cache* currently is empty.

After the first iteration, the LLM will generate a token for each query and update *KV_Cache*. For example, in the GPT-family models, *KV_Cache* is a tensor with dimensions of [*layer*, 2, *batch_size*, *mul_head*, *seq_length*, *embed_length*], where *layer* identifies the number of transformer layers of the model, 2 denotes the *Keys* and *Values*, *mul_head* denotes the number of multi-head, *seq_length* is the length of the sequence, and *embed_length* denotes the dimension of the tokenizer embedding. Similarly, as only the *seq_length* dimension is dynamic, the other dimensions are ignored in the description of this paper for convenience. *KV_Cache* represents the inter-relationships among existing tokens of a query, so its dimension now is $batch\_size \times l_{max}$.

In second iteration, the latest output token will be input to the model as the current *input_token*, whose dimension is *batch_size*. Simultaneously, as *attention_mask* is identifying the whole sequence, including the original query and the existing output tokens, it needs to add a column with the value of all 1 to the original *attention_mask*, indicating that the current *input_token* is not padding tokens. That is, the dimension of *attention_mask* currently is expanded to $batch\_size \times (l_{max}+1)$. After this iteration, the model

not only outputs new tokens for the third iteration, but also appends the inter-relationships among *input_token* input in the second iteration and previous tokens to *KV_Cache*, whose dimension changes to $batch\_size \times (l_{max} + 1)$. Then the third and subsequent iterations can be executed iteratively, *input_token*, *KV_Cache*, and *attention_mask* will be updated in each iteration, until the inference of this batch is completed.

During inference, the computation of queries of a batch is independent to each other, which allows to replenish a new query as soon as inference of any query is completed, like a relay race. The premise of this objective is that subsequent iterations can be executed and produce accurate results, which can be ensured through the padding operation and the attention masking mechanism respectively.

The details of BATON will be demonstrated by using Figure 3 as an example. The batch contains $query^0$, $query^1$, and $query^2$. The model outputs the latest $token_{i+1}$ at the end of the $i$-th iteration, and appends the new $K$ and $V$ values to $KV\_Cache_i$ to constitute $KV\_Cache_{i+1}$. And It would append an additional column to $attention_mask_i$ for $token_{i+1}$ to become $attention_mask_{i+1}$. However, if the corresponding output of $query^2$ in this iteration is ⟨EOS⟩, i.e., $query^2$ has completed the inference, the cumulative outputs of $query^2$ will be returned first and the *input_token*, *KV_Cache*, and *attention_mask* will be manipulated based on the newly inserted $query^3[t_0^3, t_1^3]$.

**Input_token.** Since the newly inserted query needs to complete the prefilling phase, i.e., it needs to compute the relationship between all tokens, then all tokens should be input into the model. Based on this, to align the dimensions of IT, it is necessary to pad the latest $token_{i+1}^0$ and $token_{i+1}^1$ of $query^0$ and $query^1$ to the same length as $query^3$. Finally, the padded $query^0$, $query^1$, and the entire $query^3$ are prepared as the new $input\_token'_{i+1}$ for the next iteration. The dimension of $input\_token'_{i+1}$ is $batch\_size \times l_{q3}$, where $l_{q3}$ is the length of $query^3$.

**Attention_mask.** To ensure the correct computation of $query^0$ and $query^1$, their previous *attention_mask* vectors cannot be changed and the new padding part should also be masked simultaneously,

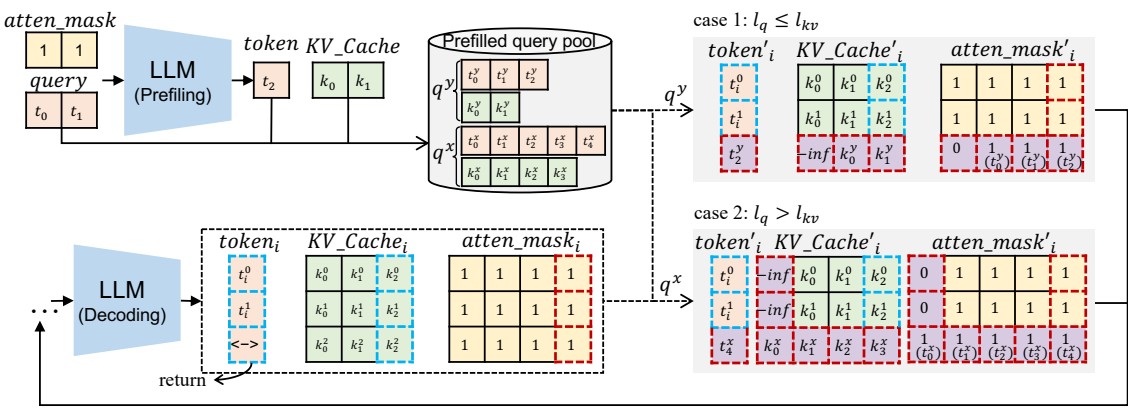

**Figure 4: Example of P&D decoupling-based vector embedding**

i.e., the *attention_mask* vectors of $query^0$ and $query^1$ are both appended with values of 0 according to the padding. For aligning the current *attention_mask* dimension, the original *attention_mask* vector of completed $query^2$ can be shaped and rewritten for reusing by the newly inserted $query^3$. Firstly, to eliminate the influence of the already cached $Keys$ and $Values$ of $query^2$ on the calculation of $query^3$, it is necessary to set all the values of the $query^2$ part of the current *attention_mask* tensor to 0. Then, the corresponding actual *attention_mask* of $query^3$, an all-1 vector with the same length of $query^3$, will be spliced on it. As a result, the dimension of $attention\_mask_{i+1}$ becomes $batch\_size \times (l_{am_i} + l_{q3} - 1)$, where $l_{am_i}$ and $l_{q3}$ is the length of $attention\_mask_i$ and $query^3$, respectively.

**KV_Cache**. First of all, maintain $Keys$ and $Values$ of $query^0$ and $query^1$ unchanged. And then, although the zeroing of *attention_mask* for $query^3$ can mask the existing $Keys$ and $Values$ of $query^2$ stored in $KV\_Cache$, it is better to set such values to negative infinity ($-inf$) to completely eliminate the impact on the calculation of $query^3$. The dimension of $KV\_Cache_{i+1}$ becomes $batch\_size \times (l_{kv_i} + 1)$, where $l_{kv_i}$ is the length of $KV\_Cahe_i$.

After the above mentioned processing, the model is able to calculate the batch with a new query properly. Regarding each query, the padding of $query^0$ and $query^1$ will not affect the output of the next iteration. The shaping of $query^3$ over the $Keys$, $Values$, and *attention_mask* of $query^2$ only results in a dimensional change actually, which can be interpreted as a kind of padding with the entire length of $query^2$ in front of $query^3$, without affecting the output result of next iteration either.

However, so far, it is impossible for all queries in an iteration to finish inference simultaneously, making it difficult to uniformly release resources occupied by $KV\_Cache$ etc., as is done in traditional batch-wise inference, and then start the inference for the next batch. If inference process is conducted according to the aforementioned way, it will lead to continual expansion in the length of $KV\_Cache$ etc., resulting in resource wastage and even potential GPU memory overflow.

Following the aforementioned way, *input_token*, $KV\_Cache$, and $attention_mask$ can not only meet the dimensional alignment requirements but also do not affect the inference results of all current processing queries and the new one.

**Resources releasing**. Essentially, it acts as padding in front of the corresponding $KV\_Cache$ and *attention_mask* vectors for each newly inserted query, which serves no purpose except placeholding. Therefore, once each query in the batch has been roundly updated, there will be certain placeholder padding in $KV\_Cache$ and *attention_mask* vectors of each query. To prevent these tensors from continuously expanding, the front overlapped placeholders of $KV\_Cache$ and $attention_mask$ can be released. This operation can be performed before inserting a new query. It is possible to assign an identifier, *index*, to each query in a processing batch, which marks the end of the padding. Consequently, the $[0 : min(index_i)]$ segments of $KV\_Cache$ and *attention_mask* should be released.

## 4.2 Vector Embedding

It is possible to achieve a query-level seamless batch-wise LLM inference through vector shaping, which involves placing placeholders in front of the newly inserted query and padding behind the currently processing queries. The first iteration following the insertion of a new query involves performing a prefill computation for it, with overhead proportional to the sequence length of the new query. Given that inference is performed uniformly in a batch, even other queries that are already in the decoding phase will also have the same overhead in this iteration, but without producing any additional and useful computational result.

Consequently, if the new query is particularly lengthy, it will distinctly decrease the computational efficiency of other queries by imposing such additional prefill computation overhead. Furthermore, after such idle computations, the $KV\_Cache$ of the processing queries will be appended with the values representing the relationship strength between all padding symbols and other existing tokens. The appended length is equal to the padding length introduced by the insertion of the new query, although these values do not contribute to the inference beyond serving as placeholders. Overall, the longer the new query, the greater the impact on the computational efficiency of the processing queries and the more unnecessary GPU resources are consumed.

If a newly inserted query contains only one token requiring computation, it will not trigger the aforementioned issues. Indeed, processing a single token is sufficient for any query during the decoding phase. To address this, Baton implements a prefilling

and decoding phases decoupling strategy. Specifically, all original queries awaiting processing are initially prefilled by the model and termed prefilled_queries, whose $KV\_Cache$ is no longer empty. It is possible to embed corresponding $Keys$ and $Values$ into the $KV\_Cache$, which allows the model to execute subsequent inference iterations based solely on the latest single token for the new query. In this context, when inserting any prefilled query into the processing batch, the padding operations for vector alignment in existing queries can be eliminated, as the input length of existing queries and the new query is uniformly 1.

Commonly, embedding the $Keys$ and $Values$ of the new query into the $KV\_Cache$ will not incur additional resource consumption, as this part of the $KV\_Cache$ would otherwise be occupied by placeholders ($-inf$) in the vector shaping scheme. It only requires additional space resources if the sequence length of the $KV$ exceeds the $KV\_Cache$'s. Assuming the length of a new query's $Key$ and $Value$ is $l_q$, and the current $KV\_Cache$ length is $l_{kv}$, these two cases of vector embedding as shown in the Figure 4.

First, if the $KV$ length of the new query is less than the current $KV\_Cache$ length, i.e., $l_q \leq l_{kv}$, the $KV$ values are embedded into $KV\_Cache$ in an end-aligned manner, and the remaining front part of $KV\_Cache$, with length $l_{kv} - l_q$, will be filled with $-inf$ for placeholding. In this situation, it allows the new query to be embedded without any additional resource consumption compared to traditional batch-wise inference. Second, if the $KV$ length of the new query exceeds the current $KV\_Cache$ length, i.e., $l_q > l_{kv}$. Since, it is necessary to save all the $KV$ values into $KV\_Cache$ to ensure accurate computation of the new query, the length of $KV\_Cache$ should be expanded to $l_q$ firstly, i.e., the length of $l_q - l_{kv}$ should be added to the left side of $KV\_Cache$. Then the expanded part will be filled $-inf$ according to vector shaping and $KV$ can be embedded into $KV\_Cache$ directly. Meanwhile, the $attention\_mask$ also should be expanded accordingly. The expansions of existing queries will be filled with 0, and, similarly, the values of new query's $attention\_mask$ will be filled with 1. At this point, the model can perform decoding computation directly for all queries of the batch according to updated $input\_token$, $attention\_mask$, and $KV\_Cache$.

### 4.3 Additional functionalities

**Preemptive scheduling**. To enhance user experience, many LLM inference servings establish Service Level Agreement (SLA) for user's query, prioritizing queries for processing. In an online inference service, a high-priority query may arise at any moment. If the inference engine is processing a batch with extensive long queries, the existing run-to-completion policy would require the high-priority query to wait until the entire batch is completed. Although BATON has enabled the execution of inference as soon as any query within the batch completes, delays remain, potentially leading to service quality degradation or failure. Fundamentally, BATON supports preemptive query scheduling. Specifically, it can temporarily store the $Keys$ and $Values$ of the batch's lowest-priority query, $q_i$, and then insert the high-priority query into the batch. Once any query of the batch is completed, the interrupted $q_i$ can be promptly re-inserted.

**Batch size scaling**. During the inference process, the resources occupied by the service continuously increase due to the expanding size of the involved $Keys$ and $Values$. Since the maximum size of required memory cannot be predicted, there is a risk of GPU memory overflow during inference, which can significantly reduce inference efficiency. As mentioned, BATON supports the interruption and resumption of queries. Therefore, it is feasible to monitor GPU resource usage in real-time. If resource utilization exceeds a predefined threshold, some queries of the processing batch can be moved to the host memory then the corresponding occupied resources can be released, allowing for flexible adjustment of the batch size during inference. Similarly, the batch size can be scaled up by inserting additional queries into the batch.

## 5 Evaluation

### 5.1 Experimental setup

**Selected model**: We utilized the llama2-7b-chat-hf model [31], which is part of the LLaMA family, specifically designed for chat-based tasks. It contains 7 billion parameters, designed to handle a wide range of natural language understanding and generation tasks. In this experiment, the model was loaded with 16-bit floating-point precision, which balances memory usage and computational efficiency, allowing for faster inference without sacrificing significant accuracy.

**Inference setup**: The key feature of our inference setup is that no sampling was applied during generation. This means that each request results in deterministic and repeatable outputs, ensuring that the generated content remains consistent for the same input across different runs. This approach is critical for fair comparison between different algorithms, as it ensures that all algorithms generate outputs of identical lengths.

**Dataset**: The proposed BATON will be evaluated from both an overall and a detailed perspective, termed *Target-1* and *Target-2*, respectively. For *Target-1*, we aim to compare ours and existing methods in terms of the query processing throughput. For *Target-2*, we want to analyze detail performance changes during inference. Due to the lack of real-world query trace, we generated two sets of simulated datasets based on task requirements: (1) A set of 120 queries, which primarily consisting of three types of queries, long input-short output, short input-long output, and short input-short output (in a ratio of 1:1:2). These correspond to common tasks such as text summarization, article generation, and general question-answering tasks like translation. The length for 'long' ranges around 4,000 words, while 'short' ranges between a few dozen to 400 words. (2) A set of 30 queries. It consists of all short input-short output, where both the input and response range from a few dozen to 200 words.

**Hardware environment**: For overall evaluation (Target-1), due to the large GPU memory demands of processing long sequence queries, we used an NVIDIA A6000 GPU, which has 48 GB of memory. For detailed evaluation (Target-2), we employed an NVIDIA 4090 GPU, which has 24 GB of memory but offers more efficient computation ability. To ensure that all comparison methods used the same resources, we applied asynchronous P&D decoupling in the experiments.

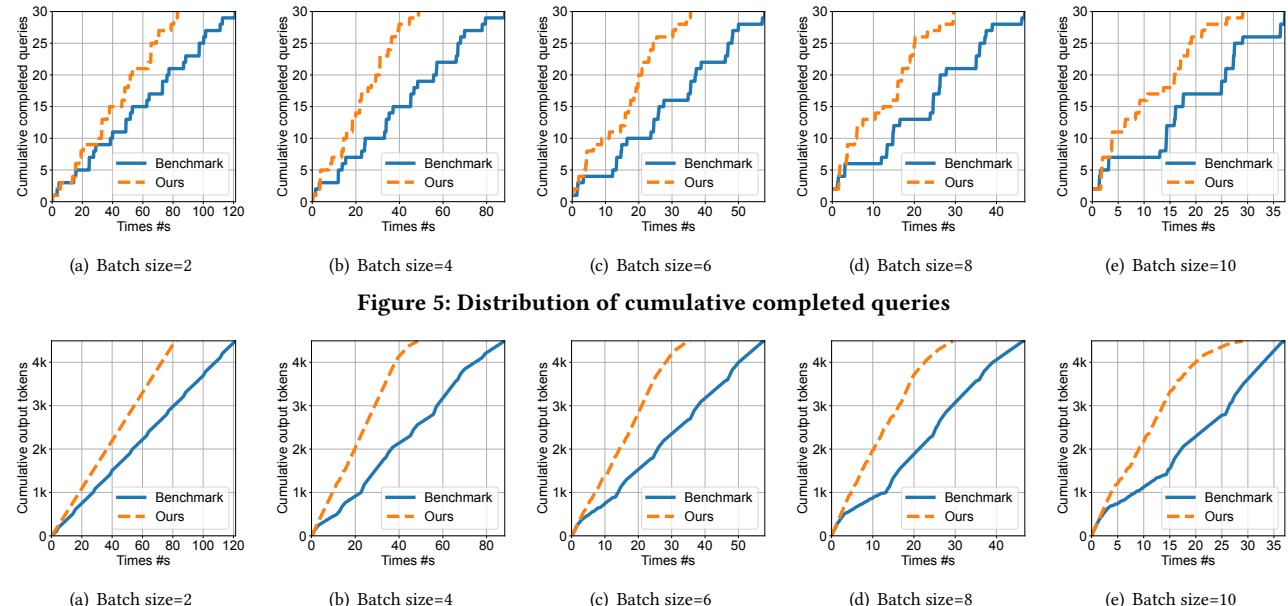

(a) Batch size=2    (b) Batch size=4    (c) Batch size=6    (d) Batch size=8    (e) Batch size=10

Figure 5: Distribution of cumulative completed queries

(a) Batch size=2    (b) Batch size=4    (c) Batch size=6    (d) Batch size=8    (e) Batch size=10

Figure 6: Distribution of cumulative output tokens

Table 1: Queries completion time (s)

| Batch size | Benchmark | PD | BATON (Ours) | BATON-PD (Ours-PD) |
|---|---|---|---|---|
| 4 | 21,901 | 21,781 | 20,549 | 11,771 |
| 8 | 6,619 | 6,577 | 3,424 | 2,654 |

Table 2: Queries completion time (s)

| Batch size | Benchmark | BATON (Ours) | Throughput imprv. |
|---|---|---|---|
| 2 | 161.77 | 83.43 | 1.94× |
| 4 | 92.68 | 49.61 | 1.89× |
| 6 | 61.70 | 37.25 | 1.66× |
| 8 | 50.07 | 31.08 | 1.61× |
| 10 | 40.45 | 30.99 | 1.31× |

**Comparison methods description**: It involves four methods in the experiment: 1) The first is the most widely used baseline method, which is the batch-wise strategy provided by the transformers [32], referred to as the 'Benchmark'; 2) The second is the P&D decoupling-based inference strategy, where in the prefill phase, queries with similar sequence lengths will be grouped into a batch, and batches are randomly combined during the decoding phase. This method is called 'PD'; 3) The third is our proposed method without P&D strategy, referred to as BATON, which is equivalent to Orca [19] in terms of throughput efficiency, but without introducing additional model parameters. 4) The fourth is BATON integrated with P&D, referred to as BATON-PD. To ensure fairness, both Benchmark and PD methods also adopt the strategy of returning a query's response immediately after its inference is complete rather than waiting for the entire batch to finish.

## 5.2 Results and analyses

*5.2.1 Target-1: overall evaluation.* The throughput of an LLM service in handling queries is a critical performance metric. We measured the completion time of first dataset processed by above four methods Benchmark, PD, BATON without P&D decoupling strategy (Ours), and BATON integrated with P&D decoupling (Ours-PD). The batch size was set to 4 and 8. To thoroughly evaluate the inference performance, all query sequence were duplicated in the experiment with batch size of 4, e.i., doubling the length of each sequence.

The results are shown in Table 1, and its shows that our proposed scheme has a clear advantage in improving the throughput of the inference system. Compared to the Benchmark (PD), our approach reduces the processing time by 46.25% (45.98%) for batch of 4, 59.90% (59.64%) for batch size of 8, respectively. Furthermore, it can be observed that there is a significant difference in the performance of BATON with or without the P&D decoupling when batch sizes of 4 and 8. By analyzing the inference logs, we noticed that doubling the sequence length will aggravate the cumulative impact of the additional prefilling computation overhead. That is, compared to the existing state-of-the-art method Orca, which lacks the P&D decoupling strategy, our method can improve the throughput by **1.29-1.75 ×**.

*5.2.2 Target-2: detailed analyses.* To further analyze the perform details of proposed scheme, we conducted inference for the second dataset, which exclusively excludes long sequences queries. This set of experiments focuses on analyzing the detail performances during inference, so Benchmark and BATON were chosen for comparison.

1) **Query processing throughput**:

•*Queries completion time.* We compared the queries completion time between BATON and the benchmark under the second query set, following the experimental setup description. As illustrated in Table 2, compared to the benchmark, BATON speeds the throughput by 1.94×, 1.89×, 1.66×, 1.61×, and 1.31× when the batch sizes are set to 2, 4, 6, 8, and 10, respectively. In this set of experiments, the throughput improvement diminishes as the batch size increases.

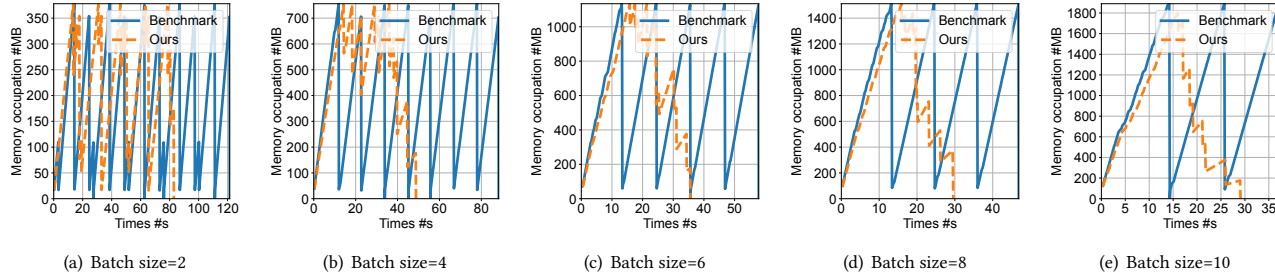

(a) Batch size=2     (b) Batch size=4     (c) Batch size=6     (d) Batch size=8     (e) Batch size=10

Figure 7: GPU memory usage of KV_Cache

This is because, during the later stage of inference, there are no additional queries to replenish the processing batch, meaning the number of effective queries being processed within the batch gradually decreases in this stage until all queries are completed. However, if the processed queries is replenished continuously, the throughput improvement will not decline with the increase in batch size.

•*Cumulative completed queries*. All methods employs the scheme that the result of a query will be returned immediately when its inference is completed, rather than awaiting the completion of all queries of the batch to return all results synchronously. The distributions of cumulative completed queries with batch size setting to 2, 4, 6, 8, and 10 are shown in the Figure 5. It can be observed that the step-like pattern of Baton is less noticeable compared to the Benchmark. This is because that the batch inference can be delayed by a query that requires more iterations in Benchmark, and this effect becomes more pronounced as the batch size increases. In contrast, Baton allows for the continuous updating of batch queries without interruption, naturally avoiding this issue.

•*Cumulative output tokens*. In a similar manner, as shown in Figure 6, we also tracked the cumulative distribution of output tokens. The benchmark method shows a distinctive multi-phase pattern, where each phase follows a similar mode: an initial linear output speed, followed by a gradual slowdown. It is also introduced by the delayed issue described above. Contrastively, the token output starts with a linear rate and then in the later stage, when no new queries are available to replenish into the processing batch, the output rate decreases gradually.

2) **GPU memory usage**. For a given model, the memory consumed by its parameters remains constant during inference. In LLMs, the *KV_Cache* represents a significant portion of memory usage, and its size grows linearly with the iterations of inference. To highlight the differences in memory consumption between Baton and the comparison methods, we measured only the memory used by the *KV_Cache*, excluding the memory occupied by model parameters.

As illustrated in the Figure 7, the benchmark method shows a clear sawtooth pattern w.r.t memory usage. This occurs because, even though individual queries within the batch complete their inference, the corresponding *KV_Cache* continues to grow until the entire batch finishes, at which point all the memory will be released. In contrast, Baton avoids releasing memory for the entire batch synchronously, instead releasing memory as outlined in subsection 4.1. Additionally, it can be observed that the peak memory usage of Baton and the benchmark method is comparable during entire inference, Baton maintains a consistently higher utilization rate, indicating more efficient resource usage.

## 6 Related Work

**LLM as-a-service**. In recent years, technologies associated with large language models (LLMs) have undergone rapid development [14, 33, 34]. These models, through adaptive instruction tuning [35], can fulfill human requirements and are available as a service to users. For instance, GPT, Llama, PaLM, ERNIE, Qwen, etc. have been effectively deployed in the cloud to provide LLM services [5, 8, 9, 26, 36], which handle vast numbers of query inferences per day. In this context, enhancing service quality and reducing inference overhead have emerged as critical research directions [19, 37].

**Efficient inference of LLM**. Efficient inference not only ensures high-quality user services but also reduces operational costs for providers [38, 39]. The technologies involved can be categorized as follows: a) Kernel Customization [40, 41]. For example, [42] reduces the need for large contiguous memory by segmenting the input vector and calculating the attention weights for each segment independently. b) Parallel Computing [43–45]. The pipeline and tensor parallelism techniques facilitate efficient multi-GPU parallel processing. c) Quantization is also an essential technology that can optimize inference processes [46, 47]. d) Some studies try to enhance batch processing during inference [48, 49], e.g., [50] groups queries based on the lengths of input vectors. However, these approaches are primarily designed for single computing engine scenarios. Baton proposed in this paper is designed for services deployed across multiple clouds and operates orthogonally to the strategies mentioned previously, allowing for integration with them.

## 7 Conclusion

In this paper, we propose a efficient batch-wise LLM inference scheme, Baton, which enables removing completed queries from or inserting new queries to the current processing batch with near-zero idle computations. We build a prototype of Baton and execute extensive experiments. Compared to the state-of-the-art solution, Baton exhibits 1.29-1.75× improvements in terms of query processing throughput.

In future work, we aim to validate the proposed scheme on the basis of real query traces, including dynamic batch size scaling and preemptive scheduling. As well as migrating it to different inference frameworks to achieve more comprehensive experimental evaluation.

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
