# OpenReview forum: "BATON: Enhancing Batch-wise Inference Efficiency for Large Language Models via Dynamic Re-batching"
_ACM.org/TheWebConf/2025/Conference — WWW 2025 Poster_

### Official Review · Reviewer_ZJJN · 2024-11-09

**Novelty:** 3
**Technical Quality:** 4

**Review:**

This paper propose an batch-wise LLM inference scheme, BATON, by dynamically adjusting processing batch. The authoes calims that the proposed method can achieve near-zero idle computations without incurring additional resource consumption. The experimental results show that compared to the only state-of-the-art solution Orca, the throughput improves by up to 1.75 times.

The Major contributions and Pros:
1. Designing a tensor shaping and embedding strategy to achieve query-level seamless batch-wise LLM inference, supporting query exits from and inserts to the current processing batch.
2. Designing  a tensor alignment policy based on P&D decoupling, avoiding the resource and computational bubbles introduced by the embedding process and freeing the implicit constraints of batch composing.
3. Conducting experiments with representative LLMs, and the results show that Baton outperforms the state-of-the-art solution w.r.t. query processing throughput by 1.29-1.75×.
4. The problem addressed by the paper is significant, as we know that the throughput performance bottleneck of LLMs is a major challenge. The author’s approach to addressing the timing issue is a good start for this field. The figures provided in the paper are well-presented, and the writing reflects the author’s solid foundation to some extent.


The method proposed by the author has achieved SOTA compared to the baseline; however, I still have the following cons:
1. The author only compared one baseline (published in 2022) and claimed it to be the latest SOTA. This is insufficient to experimentally demonstrate the advancement of BATON. It is recommended that the author review the latest literature, especially from 2023 and 2024, and include more baselines.
2. The related work section is insufficient and needs to be supplemented.
3. Evaluating the experiment using only the LLMA family is insufficient. Is this setup due to the characteristics of LLMA? Can BATON perform better on ChatGPT or the Qwen series? Please provide more experimental evidence and analysis.
4. I would like to know the advantages of the proposed solution in terms of resource utilization, including CPU and CUDA memory. This would help better understand the throughput advantages of BATON. Besides resource consumption, I would also like to know if the proposed method affects other metrics, such as inference precision and accuracy.

**Questions:**

My concerns mainly focus on the inadequacy of experimental validation. If the author can provide better explanations or additional data, this would make the work more comprehensive. As it stands, there is still significant room for improvement.

**Reviewer Confidence:**

3: The reviewer is confident but not certain that the evaluation is correct

**Scope:**

3: The work is somewhat relevant to the Web and to the track, and is of narrow interest to a sub-community

---

### Official Review · Reviewer_o929 · 2024-11-25

**Novelty:** 6
**Technical Quality:** 6

**Review:**

This paper proposes Baton, an innovative scheme, designed to enhance the efficiency of batch-wise inference for Large Language Models. It addresses the challenge of varying iteration requirements for different queries in traditional batch inference processes, which can lead to idle computations and reduced efficiency. Baton introduces a dynamic re-batching mechanism that allows for the insertion of new queries and the removal of completed ones from the current processing batch with minimal idle computation, thus optimizing resource use.

Strong Points:
SP1: The proposed Baton is a novel and efficient framework for dynamic re-batching of LLM.
SP2: The paper is written clearly and well-organized.
SP3: Baton is designed to be a generic solution that can be applied to any existing LLM using the KV-Cache policy, without the need for invasive model modifications.

Weak points:

WP1: Lack of experimental validation on the universal applicability of Baton across different LLMs.

WP2: Limited analysis on latency.

WP3:Lack of analysis for distributed systems.

**Questions:**

D1: The paper claims Baton can be applied to any existing LLMs. To demonstrate the universality of this method, more experiments on different LLMs with KV-cache policy are required.

D2: While Baton significantly improves the throughput of LLM inference by dynamically adjusting batches and reducing idle computations, the paper does not provide an in-depth analysis of how these improvements directly translate to latency reduction for individual queries.

D3: The paper does not explicitly discuss application of Baton in large-scale distributed systems. Distributed systems present unique challenges such as network latency, data synchronization, and load balancing across multiple nodes, which may affect the effectiveness of Baton's dynamic re-batching strategy. It’s an interesting research point for Baton’s future work.

**Reviewer Confidence:**

3: The reviewer is confident but not certain that the evaluation is correct

**Scope:**

4: The work is relevant to the Web and to the track, and is of broad interest to the community

---

### Official Review · Reviewer_xBeH · 2024-12-02

**Novelty:** 4
**Technical Quality:** 5

**Review:**

The paper introduces Baton, a novel approach designed to enhance the efficiency of batch-wise inference for LLMs. The method addresses the issue of inefficiencies that arise due to idle computations and resource wastage during the pre-filling process in batch inference. Baton aims to dynamically adjust the batch processing to minimize idle computations while keeping inference correct and resource-efficient.

S1: Baton introduces a novel way to manage batch-wise inference, reducing idle computations and improving overall efficiency without additional resource usage, which is a critical advancement in the field of LLM inference.

S2: The problem addressed by the paper is highly relevant to current AI models like GPT-3, where efficient inference processing is a major challenge in large-scale deployments (e.g., for web-based applications like ChatGPT).

S3: The paper presents its methodology clearly, explaining both the vector shaping and KV-cache embedding techniques in a step-by-step manner, allowing readers to understand the core process behind Baton.

W1: The paper focuses on batch-wise efficiency but does not discuss in depth how Baton scales to extremely large models or edge cases, such as when query size or batch size varies significantly. Exploring these aspects would strengthen the practical relevance of the method.

W2:  The paper mentions theoretical improvements in efficiency but does not fully explain the real-world applicability of Baton, particularly in diverse production environments.

D1: The experimental section would benefit from more details on the test setup, such as the exact dataset used, the number of queries processed, and the hardware configuration. This would allow readers to better evaluate the results' validity and reproducibility.

D2: The paper states that Baton eliminates idle computations and reduces resource usage but does not fully quantify these benefits. Including specific metrics such as time savings, memory usage, and energy consumption would enhance the impact of the findings.

**Questions:**

See Review

**Reviewer Confidence:**

2: The reviewer is willing to defend the evaluation, but it is likely that the reviewer did not understand parts of the paper

**Scope:**

4: The work is relevant to the Web and to the track, and is of broad interest to the community

---

### Official Review · Reviewer_Fabn · 2024-12-03

**Novelty:** 7
**Technical Quality:** 7

**Review:**

The paper introduces Baton, an efficient batch-wise inference scheme for Large Language Models (LLMs), addressing efficiency challenges in traditional batch-wise inference. Baton dynamically adjusts processing batches to achieve near-zero idle computations without additional resource consumption. It aligns dimensions of vectors for new queries and generates new attention masks to ensure correctness without extra resources. Baton also embeds prefilled keys and values into the KV_Cache, eliminating idle computations. Experimental results show that Baton improves the throughput of Orca by up to 1.75×.

Strengths:
1.	This paper studies an interesting and practical problem, i.e., efficient batching for LLM inference serving.
2.	This paper is well written and organized, it is easy to follow.
3.	The proposed solution which integrates vector shaping and vector embedding sounds new and promising.
4.	Extensive test-bed experiments are conducted to verify the performance of the presented system.

Weakness:
See the detailed questions.

**Questions:**

Questions:
1.	When summarizing the disadvantages of existing LLM inference serving system, Orca and FastServe are selected as the SOTA systems. However, Orca was published in 2022. My recent inference systems, such as vLLM, Distserve and Splitwise can be discussed.
2.	This paper summarized two disadvantages of existing LLM inference serving system, i.e., replicating manner makes the model consume more resources and the synchronization requirement of the linear layer computation makes the Prefilling of the new query choke the decoding of the other queries of the batch. These conclusions are not intuitive, more discussions of empirical measurements are required to better interpret the insights.
3.	The implementation details of Baton have not been discussed.
4.	In the related work section, the literature survey does not adequately cover the most recent developments in batching inference for LLMs, suggesting a need for an update. This gap may exist due to the rapidly evolving nature of the field.

**Reviewer Confidence:**

3: The reviewer is confident but not certain that the evaluation is correct

**Scope:**

4: The work is relevant to the Web and to the track, and is of broad interest to the community